# Validation of a Commercial Enzyme-Linked Immunosorbent Assay for Allopregnanolone in the Saliva of Healthy Pregnant Women

**DOI:** 10.3390/biom12101381

**Published:** 2022-09-27

**Authors:** Maria Katharina Grötsch, Denise Margret Wietor, Timm Hettich, Ulrike Ehlert

**Affiliations:** 1Department of Clinical Psychology and Psychotherapy, University of Zurich, 8050 Zurich, Switzerland; 2Department of Chemistry and Bioanalytics, University of Applied Sciences and Arts Northwestern Switzerland, 4132 Muttenz, Switzerland

**Keywords:** ELISA, salivary bioscience, allopregnanolone, reproductive mood disorders, pregnancy, validation protocol

## Abstract

Enzyme-linked immunosorbent assays (ELISAs) for saliva are simple, non-invasive methods for hormone detection. Allopregnanolone (ALLO) is a neuroactive steroid hormone that plays a crucial role in the aetiology of reproductive mood disorders. To better understand the relationship between ALLO and mood, a validated method to measure peripheral hormone levels is required. Currently, there is no commercially available ELISA with which to measure ALLO in saliva. We validated two ELISAs, developed for use with blood, with the saliva samples of 25 pregnant women, examining the range and sensitivity, intra- and inter-assay precision, parallelism, linearity of dilution, and recovery. The samples were simultaneously analysed using the liquid-chromatography–mass-spectrometry (LC-MS) method. The kits differed in range (31.2–2000 pg/mL vs. 1.6–100 ng/mL) and sensitivity (<9.5 pg/mL vs. 0.9 ng/mL), with the latter showing significant matrix effects and the former fulfilling the acceptance criteria of all the parameters. The concentrations measured with LC–MS were below the lower limit of quantification (<1.0 ng/mL) and no signal was detected. One of the tested ELISAs is a valid method for detecting ALLO in the saliva of pregnant women. It has a suitable measurement range and higher sensitivity than the conventional LC–MS method.

## 1. Introduction

The interest in quantifying steroid hormones in human, plant, and animal tissue began around 1920, and analytical methods to achieve this have been continuously developed [1]. Since their development in 1971, enzyme-linked immunosorbent assays (ELISAs) have been widely used for the detection of specific analytes [2,3]. Despite the rapidly evolving use of molecular methods, ELISAs still form the backbone of medical diagnostics, industrial practice, and scientific research [4]. They are easy to perform, safe, highly efficient, and cost-effective, and they offer high specificity and sensitivity for a variety of analytes, including steroid hormones [2,5,6]. Originally developed for the detection of immunoglobulin G (IgG) in blood serum [2], ELISAs have since been continuously adapted for use with other biospecimens, including blood plasma, urine and faeces, breast milk, cerebrospinal fluid, and saliva [7,8]. Nevertheless, methods for the quantification of steroid hormones still pose a challenge, and there is a lack of validation studies investigating standardised validation parameters and protocols [3,9]. 

Over the last four decades, salivary bioscience in particular has attracted significant of interest in medical and biopsychological research due to its advantages over blood measurements, specifically for self-administered and continuous measurement and for research on vulnerable populations [10,11,12]. Saliva collection is non-invasive, stress-free, and quick. Unlike venous blood collection, it does not require trained staff. Furthermore, following instruction, it can be performed by patients, at home. Therefore, the use of saliva as a biospecimen is an attractive and low-risk method of generating biological data.

Allopregnanolone (ALLO) is a neuroactive steroid hormone that has been shown to play a crucial role in the aetiology of reproductive mood disorders in women, such as premenstrual dysphoric disorder and peripartum depression [13,14]. It is a potent positive allosteric modulator of the gamma-aminobutyric acid (GABA) receptors, and works by potentiating the GABAa receptor function. ALLO is associated with anxiolytic, antidepressant, and stress-buffering effects in humans, but ALLO concentrations have also been found to be related to irritation, aggression, or sedation in a U-shaped manner [15]. In 2019, the Food and Drug Administration approved the intravenous injection of brexanolone, an exogenous analogue of allopregnanolone, as the first drug to specifically treat patients with postpartum depression [14,16]. Currently, zuranolone, as another analogue, is undergoing testing for the oral treatment of postpartum depression [17]. Therefore, the aim of this pilot study is to thoroughly validate two commercially available ELISA kits, which were originally developed for use with blood serum, on saliva samples. Through this, we wish to fulfil the need for a standardised validation protocol for commercially available ELISA kits for the quantification of steroid hormones.

The measurement of ALLO in saliva poses several specific challenges, including low peripheral hormone levels and, hence, low assay detection limits [11]. For the present validation study, we therefore chose pregnant women as a sample population. During pregnancy, circulating ALLO concentrations rise dramatically, reaching a six-to-tenfold increase compared to non-pregnant luteal levels [10,18,19]. The elevated circulating hormone levels during pregnancy should improve the detectability of ALLO and, thus, facilitate its measurement in saliva [20,21]. A further challenge lies in the matrix composition of saliva and other biological fluids and its effects on analyte detection [9,11]. To ensure that the change in sample matrix does not significantly affect analyte detection, parameters such as parallelism, linearity of dilution, and recovery are calculated [22,23,24,25]. Moreover, as general quality-control data, the intra- and inter-assay precision and sensitivity of the assay are determined [22,25]. In order to compare the tested ELISAs to a reference method, samples from the same subjects were simultaneously run with a liquid-chromatography–mass-spectrometry (LC-MS) method in an independent laboratory. Next to ELISA, LC–MS is one of the most widespread and sensitive methods for hormone analysis in different biospecimens [1,3]. Nevertheless, LC–MS analyses require expensive material and trained personnel. The determination of the extent to which hormone concentrations in the saliva samples of one subject correlate between ELISA and LC-MS yields important information on the validity of ELISAs in studies conducting steroid hormone analyses [26,27]. To the best of our knowledge, the present pilot study is the first to validate commercially available ELISA kits, developed for use with blood samples, for the measurement of salivary ALLO, and the findings should ensure good quality control for the tested ELISAs for application in further research.

## 2. Materials and Methods

### 2.1. Study Design and Procedure

This pilot study was an exploratory, single-centre validation study. Participants were recruited as a convenience sample through social media and provided written informed consent. Participants were compensated with a baby product worth 5 CHF. Saliva was collected once via passive drool using SaliCaps (IBL International GMBH, Hamburg, Germany). Each woman provided between 4 mL and 8 mL of unstimulated whole saliva. Participants were asked to refrain from eating, drinking, and brushing their teeth for at least one hour prior to sample collection. Samples were visually screened for contamination and excluded in the case of visible blood stains or food contamination, and were immediately frozen and stored at −72 °C until analysis. Storage times varied between one and sixteen months. ELISA analyses were conducted at the laboratory of the Department for Clinical Psychology and Psychotherapy, University of Zurich, Switzerland. LC–MS analyses were conducted at the laboratory of the Institute of Chemistry and Bioanalytics at the University of Applied Sciences and Arts, Northwestern Switzerland.

### 2.2. Participants

Participants were physically and mentally healthy pregnant women aged between 20 and 45 years. The following self-reported exclusion criteria were applied: multifoetal gestation, conception through insemination or assisted reproductive technology, medical complications in pregnancy (e.g., hypertension, (gestational) diabetes mellitus, hyperemesis gravidarum, (pre)eclampsia, suspected foetal growth restriction, foetal structural abnormalities), medical conditions or surgical intervention that might have affected ovarian function prior to pregnancy (e.g., polycystic ovary syndrome, endometriosis), current intake of hormones, diuretics, hypertensives, or vasodilators, treatment with psychotropic substances within the three months preceding study inclusion, current psychosis, bipolar disorder, posttraumatic stress disorder, eating disorder, substance abuse or dependency, current drug use and/or smoking, alcohol intake of more than one unit per day, pre-pregnancy body-mass index (BMI) < 18 or > 30, protein-restricted diet, and regular consumption of soy products. Women were recruited in all three trimesters of pregnancy.

For the present study, saliva samples were collected once from *n* = 25 healthy pregnant women between November 2020 and July 2021. The women were in all three trimesters, between gestational weeks 10 and 41 (M = 30.5, SD = 10.3). The mean age was M = 31.6 years (SD = 3.3, MIN = 22, MAX = 38) and the mean pre-pregnancy BMI was M = 23.6 (SD = 3.6, MIN = 18.1, MAX = 30.9).

### 2.3. ELISA Kit

When the present study began, in 2019, there were few commercial kits available for the detection of human ALLO in blood plasma, serum, and other biological fluids. Currently, a kit specifically designed to detect human salivary ALLO is still lacking. The two ELISA kits used for this validation study (for characteristics, see Table 1) are manufactured by Assay Genie, an Ireland-based global life-science provider of ELISAs, assays, antibodies, and proteins that was founded in 2017. Reference ranges for the concentration of salivary ALLO in pregnant women are largely lacking. Therefore, we chose two ELISA kits (Allopregnanolone (AP) ELISA Kit, SKU: UNEB0081, and SKU: UNFI0053, Assay Genie, Ireland; hereafter called kit 1 and kit 2, respectively) with different reference ranges, in order to find the best fit for this specific sample. The manufacturer tested the kits in blood samples and provided validation criteria (see Table 1). All of the following values stem from our own validation analyses.

The ELISA procedure was performed according to the manufacturer’s technical manual and no alterations were implemented. The optical density of the plates was determined using a Tecan Infinite Plex. The concentrations were calculated using the software MagellanTM (TECAN, Version 7.3, Switzerland). 

### 2.4. Validation Parameters

In line with the literature and published guidelines, six commonly used method parameters (see Table 2) were chosen for the validation of the two ELISA kits [22,23]. This protocol meets the need for rigorous quality control and standardised validation of commercially available ELISA kits.

### 2.5. LC–MS vs. ELISA Method Comparison

To compare the tested ELISA to a reference method, samples of the same subjects were simultaneously examined using the LC–MS system in the laboratory of Chemistry and Bioanalytics at the University of Applied Sciences and Arts, Northwestern Switzerland. The LC–MS method showed a linearity range of 1.07–268 ng/mL. For the sample preparation, a liquid–liquid extraction with subsequent measurement of the samples was used. A detailed description of the LC–MS method is provided in the Appendix A. 

## 3. Results

### 3.1. Intra- and Inter-Assay Precision

We determined the intra-assay precision by testing the samples of four women in replicates of 16 each in the same assay plate. The mean intra-assay CV was 7.9% for kit 1 and 5.9% for kit 2. The inter-assay precision was also determined by the mean of sixteen replicates of the samples of four different women over four days on four separate assay plates. The mean inter-assay CV was 10.8% for kit 1 and 7.3% for kit 2.

### 3.2. Assay Range and Sensitivity

According to the manual, kit 1 has a detection range of 31.2–2000 pg/mL and a sensitivity of <9.5 pg/mL. Kit 2 has a detection range of 1.6–100 ng/mL and a sensitivity of <0.9 ng/mL. The sensitivity of the kits, calculated as the mean of the absorbance of eight blank wells plus two standard deviations, was 9.4 pg/mL for kit 1 and 8.6 ng/mL for kit 2. 

### 3.3. Parallelism

The parallelism was assessed by examining four samples diluted in parallel to the standards with a serial dilution from 1:2 to 1:128 for both kits. The dilution curves of these samples were plotted against the standard curve (see Figure 1 and Figure 2). The mean percentage parallelism for the samples was 159.4% for kit 1 and 116.9% for kit 2. If a minimum required dilution (MRD [24,28]) was considered for kit 1, which was visually determined to be at least 1:5, the mean percentage parallelism improved to 104.3%. For kit 2, no MRD was deemed necessary based on visual inspection.

### 3.4. Linearity of dilution

The linearity of the dilution was assessed by spiking four samples with a high amount of analyte (2000 pg/mL in kit 1 and 100 ng/mL in kit 2) and diluting them from 1:2 to 1:128. The samples were run in duplicate. The mean percentage linearity for these samples was 76.2% for kit 1 and 107.5% for kit 2.

### 3.5. Recovery

The recovery was assessed by spiking a known amount of analyte into four different samples, as well as standard diluent. The samples were spiked with low (25 pg/mL and 2.5 ng/mL), medium (50 pg/mL and 5 ng/mL), and high (100 pg/mL and 10 ng/mL) amounts of analyte for kits 1 and 2, respectively, and run in duplicate. The mean percentage recovery for these samples was 116.5% for kit 1 and 149.5% for kit 2. A summary of all the parameters, with the comparison with the acceptance criteria, can be found in Table 3.

### 3.6. LC-MS vs. ELISA Method Comparison

The samples from 14 participants were analysed simultaneously using the LC–MS and the two ELISAs in two independent laboratories. The LC–MS analyses revealed that the concentrations of ALLO in all 14 samples were below the limit of quantification (<1.0 ng/mL) and, furthermore, no ALLO signal was detectable. The planned correlation between the results of the two methods could not be calculated.

## 4. Discussion

As there was no validated ELISA for measuring the ALLO in saliva, we validated two competitive ELISA kits, which were originally developed for use with blood samples, in the saliva of the healthy pregnant women. All the parameters yielded results within the commonly accepted criteria [22,23,24,25], with the exception of the percentage recovery for kit 2. The present pilot study meets the need for a standardised validation protocol for commercially available ELISA kits to quantify steroid hormones.

The two kits tested are comparable in terms of the validation parameters used, and differ mainly in their respective assay ranges and sensitivities. Both kits showed excellent intra- and inter-assay precision in the saliva samples, which coincided with the validation in the blood samples from the manufacturer (see Table 1). The robust results of the parallelism tests revealed that ALLO was recognised in saliva samples in the same way as in the standards, confirming little to no matrix effects in the saliva [24,29]. The graphical observation of the parallelism tests showed that an MRD of 1:5 was suitable for the saliva samples for kit 1 to prevent matrix effects, whereas kit 2 did not require an MRD [24,28]. Both kits showed an acceptable linearity of dilution [22,24]. This indicates that the detection of ALLO in saliva is not notably affected by the dilution of the sample matrix [22,28]. The recovery experiment provided information about whether the detection of ALLO was affected by differences in the sample matrix, which was measured with spiked and neat samples. Surprisingly, the 149.5% recovery rate of kit 2 was outside of the acceptable range (80–120%), whereas kit 1 yielded good results. This implies that even though saliva does not affect the parallelism and linearity of dilution of the kit, significant matrix effects are present in the recovery of the analyte [22,24]. Therefore, to ensure valid analyte detection, the kit should be further adapted for use with saliva samples. This could be accomplished by exchanging the sample diluent with neat saliva for the dilution of samples, as well as the standards to assimilate the matrix conditions and improve recovery of the analyte [7,22,24]. Further research is necessary to test this. Given that kit 1 showed results that are all within the commonly accepted criteria, the protocol does not need to be adapted to ensure the valid detection of ALLO in the saliva of pregnant women [22]. 

As the ALLO concentrations of all the samples analysed with LC–MS were below the lower limit of quantification (<1.0 ng/mL) for this method, the intended cross-validation between the methods was not possible. One reason for this lack of detectability may be that a different form of sample preparation would have been more suitable for the detection of ALLO due to the matrix effects in saliva. Instead of liquid–liquid extraction, it would have been possible to perform a solid-phase extraction. The preliminary tests for the quality control of the sample preparation, however, demonstrated a sufficient recovery of the standard stock solution in the spiked saliva samples. Other methods of sample preparation should be tested to ascertain whether the sensitivity for measuring ALLO in saliva could be improved. 

A further reason for the lack of detectability might be that a much more sensitive LC–MS system is needed to detect the ALLO concentrations in saliva. In a recent study, Mayne et al. [30] used the LC–MS method with a linearity of 0.78–100 ng/mL for the quantification of ALLO in the blood sera of pregnant women. The authors found an average ALLO concentration of 4.4 ng/mL in gestational weeks 12 to 25 and of 7.6 ng/mL in gestational weeks 22 to 32 [30]. The concentrations of ALLO in saliva are expected to be much lower than concentrations in blood [21] and would therefore not fall into the linearity of the LC–MS method used. It may be possible that a more sensitive LC–MS system would yield better results; this should be tested further. Altogether, these results imply that ELISA kit 1 in particular shows a suitable range and high sensitivity for the valid measurement of ALLO in the saliva samples of healthy pregnant women. 

The validation of this method has several strengths. To the best of our knowledge, the present study is the first to thoroughly validate commercially available ELISA kits, developed for the measurement of ALLO in blood samples, in saliva samples from healthy pregnant women. The objective validation parameters ensured that quantitative evidence was obtained for the quality of the tested ELISA kit 1 for use with saliva samples. The simultaneous analysis of the samples with LC–MS allowed a direct comparison between the immunoassay-based method and the molecular method. The findings further enhance the quality control of the tested ELISA kit and reflect the complementary relationship between the MS and IA methods [1]. Moreover, the results underline the importance of marker-specific validations to enhance the quality of quantification methods. Altogether, kit 1 proved to be a valid, non-invasive, cost-effective, and simple method to measure ALLO concentrations in the saliva of pregnant women. 

The limitations of this study are mainly based on the exclusion of other possible validation parameters. These include selectivity (the ability of the bioanalytical method to measure and differentiate the analytes in the presence of other components) [22,24], robustness (the ability of a method to remain unaffected by small variations in its parameters) [22,24], and sample stability (the chemical stability of an analyte in a given matrix under specific conditions for given time intervals) [22,24]. The selectivity was not tested, as it is part of the method validation performed by the manufacturer and is not considered to differ significantly between different biospecimens [22]. The robustness and sample stability were not deemed pivotal for this method validation because stringent compliance with the protocol was adopted during the validation to ensure the best results. Since research on sample stability for neurosteroids is lacking, we chose a conservative approach. The samples were frozen within 24 h of collection and only thawed once, under constant conditions, to minimise the influence of the sample stability on the detection of the ALLO. Further studies should investigate the sample stability in particular in order to define guidelines for best practice in saliva-sample handling for the analysis of neurosteroid hormones.

## 5. Conclusions

The ELISA kit 1 fulfilled all the acceptance criteria for the tested validation parameters, whereas kit 2 showed significant matrix effects in the recovery experiment. Kit 1 showed a more favourable assay range and sensitivity compared to the traditional LC–MS method. The present study is therefore a stepping stone towards a valid, non-invasive, cost-effective method to measure ALLO in the saliva of pregnant women. This method facilitates the generation of reference ranges for ALLO in pregnant women, which will further improve the understanding of reproductive mood disorders.

## Figures and Tables

**Figure 1 biomolecules-12-01381-f001:**
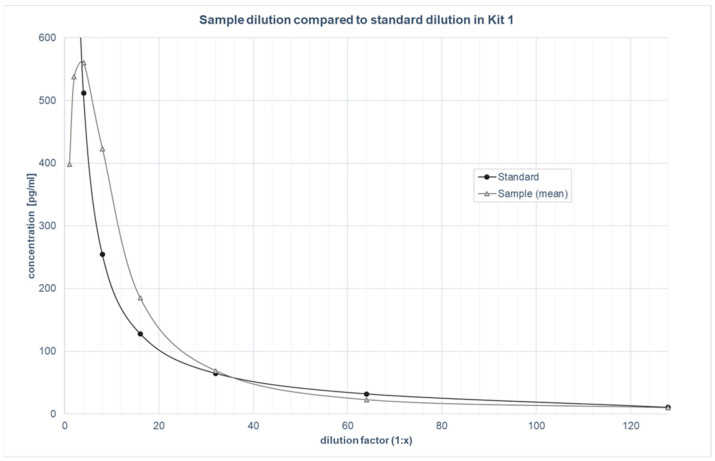
Parallelism: Sample dilution of kit 1, compared to standard dilution.

**Figure 2 biomolecules-12-01381-f002:**
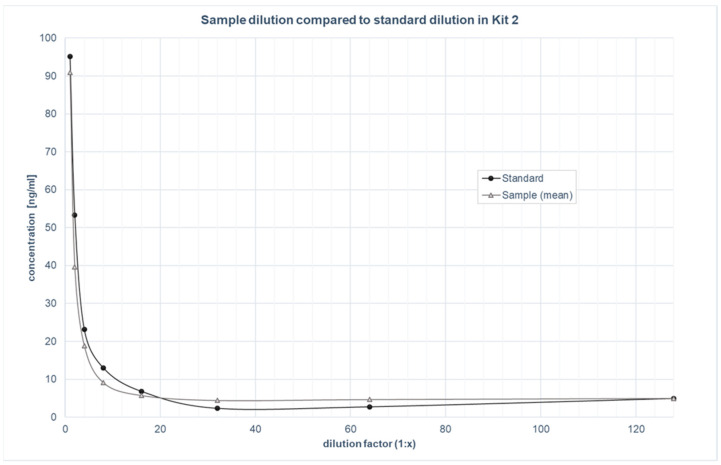
Parallelism: Sample dilution of kit 2, compared to standard dilution.

**Table 1 biomolecules-12-01381-t001:** Characteristics of the tested ELISA kits and manufacturers’ validation criteria in blood sera.

Characteristics	Kit 1 (UNEB0081)	Kit 2 (UNFI0053)
Assay range	31.2–2000 pg/mL	1.563–100 ng/mL
Sensitivity	<9.5 pg/mL	0.938 ng/mL
Assay type	competitive	competitive
Intra-assay precision	3.4%	<8%
Inter-assay precision	5.6%	<10%
Linearity of dilution	80–120%	88–105%
Recovery	102%	94%

**Table 2 biomolecules-12-01381-t002:** Validation parameters of the ELISA method.

Parameter	Description	Acceptance Criteria	Formula
Intra-assay precision	Reproducibility between wells within one assay plate	<10%	%CV = SD/M × 100
Inter-assay precision	Reproducibility between wells between assay plates, done on different days	<15%	%CV = SD/M × 100
Sensitivity	The lowest signal that can be distinguished from the background	*none*	= M_OD_ + 2 SD
Parallelism	Provides confirmation that the analyte is recognised in the natural sample in the same way as in the standards, measured with neat samples	75–125%	%Parallelism = (measured concentration/(previous measured value in the dilution series/dilution factor)) × 100
Linearity of dilution	Used to determine whether dilution of the analyte is affected by dilution of sample matrices, measured with spiked samples	70–130%	%Linearity = (measured concentration/(previous measured value in the dilution series/dilution factor)) × 100
Recovery	Used to determine whether analyte detection is affected by differences in sample matrices, measured with spiked and neat samples	80–120%	%Recovery = ((measured concentration spiked sample—measured concentration neat sample)/theoretical concentration spiked) × 100

**Table 3 biomolecules-12-01381-t003:** Summary of the results of the two validated kits for each parameter in comparison with the acceptance criteria.

Parameter	Acceptance Criteria	Kit 1 (UNEB0081)	Kit 2 (UNFI0053)
Intra-assay precision	<10%	7.9%	5.9%
Inter-assay precision	<15%	10.8%	7.3%
Parallelism	75–125%	104.3%	116.9%
Linearity of dilution	70–130%	76.2%	107.5%
Recovery	80–120%	116.5%	149.5%

## Data Availability

Data are available upon request.

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
