# Peer review of "Validation of a Commercial Enzyme-Linked Immunosorbent Assay for Allopregnanolone in the Saliva of Healthy Pregnant Women"

_biomolecules, 2022, doi:10.3390/biom12101381_

Round 1
Reviewer 1 Report
In the present pilot study, the Authors aimed to validate commercially available ELISA kits, developed for use with blood samples, for the measurement of salivary Allopregnanolone, and these findings might ensure good quality control for the tested ELISAs for application in further research.
Overall, I found the present study timely, well conducted, well written and scientifically sound. However, I have some suggestions aimed to improve the quality of the paper and these are outlined below:
1) I suggest the Authors to add a brief note in the Introduction specifying that, in 2019, the Food and Drug Administration approved brexanolone, an exogenous analog of allopregnanolone, as the first ever drug to be specifically indicated for treating patients with Post-Partum Depression, a condition characterized by high risk of suicide, with appropriate references (see dois 10.3390/diseases9030052 and 10.3389/fpsyt.2016.00138).
2) How many healthy pregnant women were screened, but refused to participate? Moreover, please note that the sample is a convenience sample.
3) Besides, was any recompense provided to the participants?
4) How "...current psychosis, bipolar disorder, post-traumatic stress disorder, eating disorder, substance abuse or dependency..." were diagnosed and how many subjects excluded?
5) Were the presence of an intellectual disability assessed and how?
6) Pre-pregnancy body mass index (BMI) was objectively or subjectively assessed?
Reviewer 2 Report
Dear Authors,
Thank you for such interesting and well-written manuscript. It is a highly important area of diagnostics and it requires accurate and precise measurment. I'd like to recomend only two minor changes:
1. Please, provide information about possibility of false-positive results for immunoassay because of "hook" effect;
2. For LC-MS/MS and GC-MS/MS few additional sentencies with references about possible derivatization techniquies and reasons for it are required.
Warm regards
Reviewer 3 Report
This paper entitled “Validation of a commercial enzyme-linked immunosorbent assay for allopregnanolone in the saliva of healthy pregnant women” by Grotsch et al. validated a commercial ELISA kit for allopregnanolone in the saliva.
This study validated a validated a commercial ELISA kit for allopregnanolone in the saliva. However, there are commercial ELISA kit for saliva available. This makes the novelty of this study very limited.
The major short of this paper is this paper only presented their final result without any measurement data. Other than that, there are some of the concerns will be addressed as follows:
1 How authors validate the assay is confusing.
2 How authors pick these two kits? Have they considered compare between different brands?
3 Line 113 authors claimed that:” To date, a kit specifically designed to detect human salivary ALLO is still lacking.” However, there are Invitrogen ELISA kit (Catalog # EIAALLO) claimed the works with saliva sample.
4 “Sample characteristics” should be in the method section.
Round 2
Reviewer 3 Report
I appreciate authors' time and effort to answer my questions. However, the main 2 question still remains.
1) As authors mentioned, many manufacturers listed saliva as one of their sample type. To my understanding, this means saliva has already been tested and verified as an acceptable sample type by the manufacturer. Authors claimed in their response: “Many manufacturers list different biospecimen in their technical guides, but do not specify if the respective kit has been tested and validated with those biospecimen.” Do they have any evidence if manufacturers never tested these sample type just list whatever they what?
2) This paper is lack of how they perform experiment and measurement data. Only result is presented.
Author Response
Response to the reviewers – round 2
I appreciate authors' time and effort to answer my questions. However, the main 2 question still
remains.
We would like to thank the reviewer for this feedback and for deepening this important issue.
We would like to take the time to elaborate on those points again.
1) As authors mentioned, many manufacturers listed saliva as one of their sample type. To my
understanding, this means saliva has already been tested and verified as an acceptable sample
type by the manufacturer. Authors claimed in their response: “Many manufacturers list different
biospecimen in their technical guides, but do not specify if the respective kit has been tested
and validated with those biospecimen.” Do they have any evidence if manufacturers never tested
these sample type just list whatever they what?
We would like to thank the reviewer for raising this point. We cannot give a general answer to
this point, but only report what we experienced in the course of this method validation. As stated
in the paper, the manufacturer of the two tested kits promotes both kits as possible to use with
saliva samples. We asked the manufacturer for the validation data with salivary samples,
whereupon they answered us, that only one of the two has been specifically validated with
salivary samples. The same has happened with another manufacturer: they stated that the kit
can be used with saliva samples, but could not provide any validation data.
Another important point here is that, even if a kit has been validated by the manufacturer, it can
still be beneficial to conduct an in-house validation to ensure that the chosen method is the best
fit for the intended use. This has also been stated in the literature: “However, the quality of
ELISA methods varies, which may introduce both systematic and random errors. This urges the
need for more rigorous control of assay performance, regardless of its use in a research setting,
in clinical routine, or drug development.” (Andreasson et al., 2015).
A third point we would like to highlight is that we validated the kit specifically with samples of
pregnant women. This has not been done before, neither by the manufacturers nor other
researchers. This is important, because physiological levels of steroid hormones are different in
pregnancy. As reliable ELISA results are highly dependent on the free levels of hormones in the
sample, we tested two different kits with different analytical ranges to ensure the best fit for the
intended use. With this, we make sure that this method is thoroughly validated not only with
salivary samples, but with salivary samples of pregnant women in all three trimesters.
Taken together, we stand by the point that the validation of a commercial ELISA kit, originally
developed for the use with blood serum, on saliva samples is necessary. With this, we wish to
fulfil the need for a standardised validation protocol for the detection of allopregnanolone in
salivary samples of pregnant women.
2) This paper is lack of how they perform experiment and measurement data. Only result is
presented.
We would like to thank the reviewer for raising this point. We thought about including raw data
into the manuscript, but decided against it. One reason is, that we analysed samples of 25
pregnant women in a total of 320 repetitions. To report all of this could have been extensive and
unmanageable. However, all data are, of course, available upon request.
Regarding the point how we performed the results, we think we reported all necessary
information to ensure this paper can be used as a protocol for future studies. We included the
formula to each parameter and the detailed description on how we conducted the analysis. The
ELISA was used according to the manufacturers protocol which can be obtained online or upon
request from the authors. The LC-MS/MS method is described in detail in the appendix. For any
specific questions about the method, the authors can be contacted at any time.
We hope that with this response we have answered any open questions. We would like to take
the opportunity again to thank the reviewer for their highly appreciated feedback to improve our
manuscript and ensure a thorough peer-review process.